# Femtosecond Laser-Induced Photothermal Effects of Ultrasmall Plasmonic Gold Nanoparticles on the Viability of Human Hepatocellular Carcinoma HepG2 Cells

**DOI:** 10.3390/cells13242139

**Published:** 2024-12-23

**Authors:** Poornima Budime Santhosh, Kamelia Hristova-Panusheva, Todor Petrov, Lyubomir Stoychev, Natalia Krasteva, Julia Genova

**Affiliations:** 1Institute of Solid State Physics, Bulgarian Academy of Sciences, Tzarigradsko Chaussee 72, 1784 Sofia, Bulgaria; poorni@issp.bas.bg (P.B.S.); petrovts@gmail.com (T.P.); stoychev@issp.bas.bg (L.S.); 2Central Laboratory of Solar Energy and New Energy Sources, Tzarigradsko Chaussee 72, 1784 Sofia, Bulgaria; 3Institute of Biophysics and Biomedical Engineering, Bulgarian Academy of Sciences, Acad. G. Bonchev Str. Bl.21, 1113 Sofia, Bulgarianatalia.krasteva@yahoo.com (N.K.); 4Faculty of Applied Mathematics and Informatics, Technical University of Sofia, 8, Kliment Ohridski St, 1000 Sofia, Bulgaria

**Keywords:** 343 nm light, femtosecond laser, gold nanoparticles, photothermal therapy, cell viability, cancer cells, hepatocellular carcinoma

## Abstract

Laser-induced photothermal therapy using gold nanoparticles (AuNPs) has emerged as a promising approach to cancer therapy. However, optimizing various laser parameters is critical for enhancing the photothermal conversion efficacy of plasmonic nanomaterials. In this regard, the present study investigates the photothermal effects of dodecanethiol-stabilized hydrophobic ultrasmall spherical AuNPs (TEM size 2.2 ± 1.1 nm), induced by a 343 nm wavelength ultrafast femtosecond-pulse laser with a low intensity (0.1 W/cm^2^) for 5 and 10 min, on the cell morphology and viability of human hepatocellular carcinoma (HepG2) cells treated in vitro. The optical microscopy images showed considerable alteration in the overall morphology of the cells treated with AuNPs and irradiated with laser light. Infrared thermometer measurements showed that the temperature of the cell medium treated with AuNPs and exposed to the laser increased steadily from 22 °C to 46 °C and 48.5 °C after 5 and 10 min, respectively. The WST-1 assay results showed a significant reduction in cell viability, demonstrating a synergistic therapeutic effect of the femtosecond laser and AuNPs on HepG2 cells. The obtained results pave the way to design a less expensive, effective, and minimally invasive photothermal approach to treat cancers with reduced side effects.

## 1. Introduction

Cancer is one of the most dreaded diseases and a leading cause of death worldwide. The prevalence of cancer is steadily increasing day by day, and the World Health Organization (WHO) has estimated over 35 million new cancer cases by 2050 [1]. Among the different types of cancer, hepatocellular carcinoma (HCC) is the most prevalent form of liver cancer in adults [2]. As per the GLOBOCAN reports, HCC accounted for 830,000 deaths in 2020 globally, ranking it as the third leading cause of cancer-related deaths [3]. To address the increasing mortality rate of HCC, there is an urgent need to gain knowledge on preventive measures, public awareness, healthy lifestyles, and improved therapeutic approaches. Though different traditional cancer therapy methods such as chemotherapy, radiation therapy, and surgery have been widely used, several disadvantages (such as tissue necrosis, incomplete destruction of cancer cells, subsequent tumor recurrence, non-specific drug release, and drastic side effects) remain as the key challenges [4,5]. Consequently, early diagnostics, smart drug delivery systems that specifically target cancer cells, and efficient therapeutic methods are crucial to overcome the existing problems. With the advent of nanotechnology, novel strategies have evolved to improve the current treatment modalities. One such approach is laser-induced photothermal therapy (PTT) using plasmonic nanoparticles [6,7].

PTT is a minimally invasive cancer therapy modality that generally employs photothermal agents to transform the absorbed light energy into heat [8]. This leads to a localized temperature increase in the targeted region, a process called photo-hyperthermia, which causes thermal damage to various cellular components, membrane rupture, and denaturation of proteins and DNA [9]. This process is also reported to trigger the production of reactive oxygen species, which cause oxidative stress that adversely affects the various biochemical pathways of cells and induces cell death mechanisms such as apoptosis and coagulative necrosis [10]. In order to destroy only the cancer cells and spare healthy neighboring cells, plasmonic nanoparticles are widely used in PTT. These particles have a unique property called localized surface plasmon resonance (LSPR), which enables them to absorb light strongly at specific wavelengths and produce localized heat in the surrounding tumor region. Among the different plasmonic nanomaterials, such as gold, platinum, and silver, gold nanoparticles (AuNPs) are a promising candidate in various biomedical applications due to their high biocompatibility, ability to conjugate with different ligands that can bind specifically to cancer cells, and high photothermal conversion efficiency [11,12,13]. By modifying their size and shape, the LSPR peaks of gold nanomaterials can be manipulated to absorb light at specific wavelengths, rendering them potential agents for various biomedical applications, including screening, drug delivery, and cancer therapy [14,15]. Therefore, based on specific requirements, surface-engineered gold nanomaterials with different morphologies have been widely synthesized to explore laser-induced PTT on various types of cancer [16,17].

Among the different types of lasers, continuous-wave (CW) or pulsed lasers are often used in PTT to induce cancer cell death [18,19]. CW lasers emit light continuously, leading to excessive heat production that might damage the healthy cells near the tumor region, which is a major drawback [20]. This problem can be overcome by using pulsed lasers, which emit laser light in short pulses with high energy peak power for a short duration, in either a nanosecond (10^−9^ s), picosecond (10^−12^ s), or femtosecond (10^−15^ s) timescale. An ultrafast fs laser system emits extremely short light pulses that can generate high energy peaks in femtosecond (fs) durations with specific time intervals and high precision, which makes them potential tools in various biomedical applications including cataract surgery, analysis of biological tissues, and imaging [21]. Due to these unique features and minimal thermal damage to healthy cells, fs laser pulses are promising tools in PTT. So far, only a few studies have investigated the influence of ultrafast pulsed fs lasers on cancer cells in detail. There is a huge gap in understanding how the surface modification of plasmonic nanomaterials and altering the laser properties can influence the differentiation and viability of cancer cells. In addition, there is room to investigate laser-induced changes in various biochemical processes and cell death pathways.

Most of the PTT techniques, including tumor ablation technology, focus on rapidly increasing the temperature surrounding the tumor area using high-intensity lasers [22]. Though this method is effective, repeated exposure to high-intensity laser light for a longer duration may be detrimental to nearby healthy tissues and include the risk of intense burns, infections, and skin cancer [23]. Therefore, it is important to choose the right type of laser with a suitable intensity, wavelength, and exposure time to minimize the side effects. In this respect, the present study aims to analyze the photothermal effects of ultrasmall dodecanethiol-stabilized AuNPs, using a 343 nm shorter-wavelength, ultrafast fs laser with low intensity (0.1 W/cm^2^) for 5 and 10 min, on the viability of human hepatocellular carcinoma (HepG2) cells treated in vitro. Due to their ultrasmall size, we anticipated that these AuNPs could easily be taken up by the cells, and that the hydrophobic dodecanethiol coating on the surface of the AuNPs would facilitate their incorporation inside the cell membranes to achieve significant results. Though various nanomaterials and laser-based studies have been previously reported, to the best of our knowledge, this is a novel study that uses hydrophobic AuNPs that can be incorporated between the lipid bilayer of the cell membrane to achieve better therapeutic results based on low-intensity fs lasers. The laser-induced temperature variations were continuously monitored throughout the experiments. The morphology and viability of HepG2 cells after laser exposure were compared to those of the control cells (untreated, non-irradiated) and cells that were treated with AuNPs but not irradiated.

## 2. Materials and Methods

### 2.1. Characterization of AuNPs

The morphology of dodecanethiol-stabilized AuNPs (NanoComposix Inc., San Diego, CA, USA) was observed under transmission electron microscopy (TEM) (model JEM-2100, JEOL Ltd., Tokyo, Japan), operated at 200 kV. The zeta potential and size distribution of the AuNPs were measured using a Zetasizer Nano ZS (Malvern Instruments, Malvern, Worcestershire, UK) at a laser wavelength of 633 nm, a scattering angle of 173°, and a temperature of 25 °C. The UV-Vis spectral analysis was conducted on a Specord 210 Plus spectrophotometer (Edition 2010, Analytik Jena AG, Thuringia, Germany).

### 2.2. Cell Culture

The HepG2 cells were cultured in Dulbecco’s modified Eagle’s medium (DMEM) supplemented with 10% fetal bovine serum (FBS; Sigma-Aldrich, Steinheim, Germany) and 1% (*v*/*v*) HEPES (Gibco™/Thermo Fisher Scientific, Waltham, MA, USA). The cells were incubated at 37 °C in a humidified atmosphere of 5% CO_2_, and the culture medium was changed once every 2 days. For the cell experiments, the cells were detached with trypsin-EDTA (Sigma-Aldrich, T4049), counted in a Neubauer hemocytometer (Feinoptik, Brokdorf, Germany), and plated at a density of 2.5 × 10^4^ cells per well in a 24-well plate. After 24 h of cultivation, the cells were treated with AuNPs at a 0.1 mg/mL concentration and incubated for 30 min before exposure to laser light. Regarding the controls, we implemented multiple experimental controls to ensure the validity of our results. Cells neither treated with AuNPs nor exposed to laser irradiation were taken as negative controls. Cells treated with AuNPs but not irradiated with the fs laser were also taken as controls. The cells not treated with AuNPs but exposed to laser irradiation at 343 nm for 5 and 10 min with a power density of 0.1 W/cm^2^ were taken as another set of controls. These controls allowed us to confirm that the observed effects on cell viability and morphology were specifically due to the synergistic interaction between AuNPs and laser irradiation.

### 2.3. Laser Setup and Laser-Induced Temperature Increase In Vitro

For the PTT experiments, a Pharos laser model Ph2-10-1000-02-H0-B (Light Conversion, UAB, Vilnius, Lithuania) operating at a 100 kHz pulse repetition rate and 130 fs pulse duration was used. A pulsed laser with a 343 nm wavelength at a low intensity (0.1 W/cm^2^) was used to irradiate the samples for 5 and 10 min. The irradiation of the samples was performed in burst mode (8 sub-pulses per pulse) at the automated harmonic generator, operating at a wavelength (λ) of 343 nm with a maximum output power of 2.9 W. The attenuation for 343 nm was 50% from the nominal power, in order to obtain a 0.1 W/cm^2^ working intensity. Laser irradiation was performed 30 min after adding the AuNPs to the wells. The temperature changes during PTT were measured at various time intervals by an infrared thermometer (Hikmicro, Hangzhou, Zhejiang, China).

### 2.4. Cell Morphology Changes

Morphological changes in the control cells (untreated and non-irradiated), cells treated with AuNPs alone (non-irradiated), and cells treated with AuNPs and laser irradiation were studied by phase-contrast microscopy. The micrographs were captured at a magnification of 10× using a Carl Zeiss microscope (Axiovert 25, Jena, Germany) equipped with a CCD camera.

### 2.5. WST-1 Cell Viability Assay

The cytotoxicity assessment of PTT was performed using the WST-1 assay (Sigma-Aldrich Co.), as previously described [24]. Initially, the cells were seeded into 24-well plates at a density of 2.5 × 10^4^ cells per well and allowed to incubate for 24 h at 37 °C and 5% CO_2_. Then, the culture medium was replaced with a fresh medium, and the cells were treated with AuNPs at a 0.1 mg/mL concentration and incubated for 30 min before irradiation. At the end of incubation, the cell medium was removed and replaced with a new one. The WST-1 reagent was added directly to the cells at a ratio of 1:10, following the manufacturer’s instructions. After 1 h of incubation at 37 °C in darkness, the amount of formazan produced by the cells was measured at 450 nm wavelength using a standard microplate reader (Thermo Scientific Multiskan Spectrum, Tokyo, Japan). From the optical density (OD) values, the cell viability percentage was calculated, and the results were graphically presented with bars denoting the mean values ± STDV of three experimental repetitions.

### 2.6. Statistical Analysis

All data are presented as the mean value ± standard deviation (SD) of three independent experiments for each sample. The significance of the statistical differences between the control and experimental groups was determined using Student’s t-test, and probability (*p*) values less than 0.05 were considered statistically significant.

## 3. Results and Discussion

### 3.1. Characterization of AuNPs

The core size of the spherical AuNPs was found to be 2.2 ± 1.1 nm using TEM (Figure 1a), and their hydrodynamic diameter was measured to be 3.5 ± 1.6 nm using DLS (Figure 1b). The zeta potential of the dodecanethiol-stabilized AuNPs was found to be −14.07 ± 0.08 mV.

Most of the previous reports on AuNP-mediated photothermal therapy employed particles whose size was usually larger than 100 nm and longer-wavelength lasers in the near-infrared (NIR) region due to their deep tissue penetration abilities and less interaction with the cellular components [25,26]. However, it is important to note that the LSPR property of gold nanomaterials is highly dependent on their morphology (size, shape) and surface characteristics. It is worth mentioning that smaller AuNPs have their peak maximum absorption (λ_max_) at shorter wavelengths, generally in the UV or visible range of the electromagnetic spectrum. Conversely, as the AuNPs’ size increases, their peak absorption maximum shifts towards longer wavelengths in the NIR range. In our case, we intended to incorporate the AuNPs inside the lipid membrane to achieve an excellent therapeutic effect. To fit inside the hydrophobic region of the membrane interior, the nanoparticles have to be coated with a hydrophobic layer and be smaller in size. Therefore, hydrophobic dodecanethiol surface-coated ultrasmall AuNPs that could be incorporated inside the cell membranes were used in this work. Reports have also shown that if the particle size is small, the cellular internalization will be more efficient [27,28]. Due to their ultrasmall size, the UV-Vis spectrum showed a sharp absorption peak around 200 nm. Hence, rather than NIR wavelengths, we chose a 343 nm shorter-wavelength laser in the ultraviolet range of the spectrum, which is closer to the maximum absorption value of these AuNPs, to achieve better photothermal conversion efficiency.

### 3.2. Laser-Induced Photothermal Effects of AuNPs

In photothermal cancer therapy, fs lasers are safe tools, as they are too short to transmit heat or trigger inflammation in biological samples [29]. In recent years, these lasers have been used to treat different types of cancer cells effectively. For instance, Liu et al. [30] reported that gold nanobipyramids irradiated with low-intensity fs laser light for a shorter exposure time exhibited excellent photothermal conversion efficiency and induced rapid cell death in human hepatocellular carcinoma (HepG2) cells. In this work, 20 s of pulsed laser irradiation with 3 mW intensity effectively induced cell death by apoptosis rather than necrosis, which significantly reduced the disruption of the neighboring normal tissues. Taha et al. [31] reported that tunable fs laser radiation at different wavelengths in the UV, visible, and NIR ranges exerted a significant anticancer effect against T47D breast cancer cells. Thogersen et al. [32] reported that high-intensity fs laser pulses at 200, 266, 400, and 800 nm significantly reduced cell viability in human cervical cancer cells. The experimental setup and schematics of our femtosecond laser system are shown in Figure 2.

As our work was focused on investigating the photothermal-induced cell death mechanism, we first analyzed the temperature increase induced by fs laser pulses in the cell culture medium (DMEM) with 10% FBS buffer. For this purpose, the cells treated with and without AuNPs were exposed to a 343 nm wavelength laser with low intensity (0.1 W/cm^2^) for 5 and 10 min. The consequent variation in the temperature of the irradiated wells was constantly analyzed using an infrared thermometer during PTT. As shown in Figure 3, the temperature of the cell medium treated with AuNPs and subsequently exposed to laser irradiation started to increase steadily from the room temperature of 22 °C and reached high temperatures of 46 °C and 48.5 °C after 5 and 10 min, respectively. Contrastingly, cells without AuNPs exposed to the same laser intensity, wavelength, and duration showed only a slight increase in temperature. This confirms that the AuNPs played a critical role in inducing the photothermal effect.

Though lasers in the NIR wavelength region are widely used in PTT due to the minimal absorption by body components in this range, studies based on using lasers in the UV and visible regions, with different intensities and exposure times, also showed a superior anticancer effect [33,34,35]. For instance, Gupta et al. [36] have reported that treating lung and breast cancer cells with superparamagnetic iron oxide (Fe_3_O_4_) nanoparticles and irradiating them with a 635 nm wavelength laser in the visible region for 17 min induced a significant rise in their temperature and reduced cell viability remarkably.

Generally, PTT employs elevated temperatures to kill tumors and induces cell death via two major pathways: either by apoptosis or necrosis [37]. Apoptosis is a programmed cell death process that eliminates unwanted and highly damaged cells beyond repair to maintain the smooth functioning of the body [38]. On the other hand, necrosis induces cell death primarily through inflammation and leads to leakage of the tissue contents. In the case of cancer cells, the leakage of cellular components can lead to cancer metastasis, paving the way for tumor recurrence, which is very dangerous [39]. Necroptosis is another mechanism of cell death, which is a regulated form of necrosis characterized by swelling of organelles and cells, cell membrane rupture, and the release of intracellular components [40,41,42,43].

Recent reports have shown that temperatures of the tumor region in the range of 43–50 °C induce the apoptosis process, and temperatures above 50 °C induce cell death via necrosis [44,45]. Zhang et al. [46] reported that PTT destroys tumors predominantly by the mechanisms of apoptosis and necroptosis at 46 °C. Hence, it is critical to increase the temperature of the tumor region to between 43 and 50 °C during PTT to induce cell death primarily by apoptosis and not necrosis, in order to reduce the damage to surrounding healthy cells during cancer treatment. In this context, our results, indicating a temperature rise to less than 50 °C, suggest that the plausible cell death mechanism could be apoptosis or necroptosis. Our results coincide with the results of Suarasan et al. [47], who reported that the application of a 785 nm wavelength laser with 2.7 W/cm^2^ intensity for 15 min on B16.F10 mouse melanoma cells treated with triangular AuNPs increased the tumor temperature from 22 °C to 47 °C, inducing cell death mainly by apoptosis.

### 3.3. Cell Viability Assays

The WST-1 assay is a widely used colorimetric test to quantify the proliferation and viability of cells [48]. As shown in Figure 4, the WST-1 assay results indicated no significant difference in cell viability between the controls (untreated and non-irradiated cells) and cells treated with AuNPs alone but not irradiated. This can be attributed to the biocompatible nature of AuNPs, which has been well reported previously [49,50,51]. On the other hand, the percentage of cell viability decreased to 60.6% and 53.8% in the case of cells treated with AuNPs and irradiated with a 343 nm laser with 0.1 W/cm^2^ intensity for 5 and 10 min, respectively. This was due to the synergistic or combined effect of AuNPs and laser light, which induced photothermia effectively to cause a significant reduction in cell viability. Our results coincide well with those of previous studies based on laser-induced photothermal studies using AuNPs [52,53].

In order to destroy most of the cancer cells, some studies have enhanced the laser power or exposure time to damage most of the cancer cells using gold nanomaterials of various anisotropic shapes, such as gold nanostars, nanorods, and nanobipyramids [54,55,56]. Among these shapes, extensive research has been performed using gold nanorods (AuNRs) to demonstrate their multifunctional potential in diverse biomedical applications, such as imaging, biosensors, and targeted cancer therapy [57,58]. A unique feature of AuNRs is that, as their aspect ratio (length/width) increases, the longitudinal plasmon resonance redshifts to longer wavelengths, which enables strong absorption of light in the NIR region [59]. This feature is beneficial for achieving a synergistic PTT effect to improve the therapeutic efficacy. For instance, El-Sayed et al. [60] compared the effects of PEG-AuNRs injected via either the intratumoral or intravenous route to Ehrlich carcinoma tumor-bearing mice every week and subsequently exposed to NIR laser irradiation at an 800 nm wavelength with a high intensity of 50 W/cm^2^ for 2 min. The tumor growth rate was monitored up to 47 days. The results showed that the tumor growth was significantly arrested in the test group treated with AuNRs and exposed to the NIR laser, whereas the size of the tumor increased considerably (about 6.3-fold compared to the initial tumor size at the beginning of the experiment) in the untreated control group. This confirms the synergistic effect of high-intensity lasers and AuNRs in PTT.

In general, the lack of precise tumor visualization techniques remains a major challenge to determine the optimal concentration of nanoparticles required for effective therapy. To overcome this issue, Parchur et al. [61] conjugated AuNRs with a magnetic resonance (MR)/X-ray contrast agent and injected them into rats bearing colorectal liver metastases. The photothermal ablation was performed with a diode fiber laser using a continuous-wave 808 nm NIR laser. After irradiation with 0.7, 1.0, and 1.5 W/cm^2^ exposures for 3 min, ΔT^max^ reached 12, 19, and 29 °C, respectively. The results showed that this approach was effective to achieve interventional image-guided PTT of tumors. To explore the multifunctional potential of AuNRs as probes for optical imaging, SERS, and PTT agents, Li et al. [62] synthesized hybrid nanoparticles (HNPs) by conjugating AuNRs with CaMoO_4_:Eu NPs and the cancer biomarker epidermal growth factor receptor. The HNPs were then treated with human lung cancer (A549) and mouse hepatocyte (AML12) cells and irradiated with an 808 nm laser at 1 W cm^−2^ power density for 5 min. The experimental results showed that the HNPs had a heat conversion efficiency of 25.6%, and a hyperthermia temperature of 42 °C was achieved by adjusting either the concentration of HNPs, the laser power, or the irradiation time. The cell viability data indicated that the group treated with HNPs showed the least percentage of viability (5 ± 2%), signaling a significant PTT effect.

Next, to visualize the morphological changes of cells under different treatment conditions, they were observed using an optical microscope (Figure 5). As shown in Figure 5a, the control (untreated, non-irradiated) cells remained intact, and their proliferation pattern was normal. Figure 5b,c depict slight morphological changes and detachment of the cells untreated with AuNPs but irradiated with the fs laser alone at a 0.1 W/cm^2^ power density for 5 min and 10 min, respectively. In the case of cells treated with AuNPs alone, no significant morphological changes were observed (Figure 5d). On the other hand, the cells treated with AuNPs and irradiated with fs laser irradiation exhibited the typical characteristics of cell death, such as cell rounding and shrinkage, loss of membrane integrity, cell detachment, and nuclear disintegration, compared to the control and non-irradiated cells.

Furthermore, the images in Figure 5e,f also depict the formation of characteristic bubbles in the cytoplasm of the cells treated with AuNPs and laser light. However, these bubbles were not observed in the non-irradiated and pure HepG2 cells taken as controls. This implies that the bubble formation was caused by the pulsed-laser-activated AuNPs. Numerous reports have suggested that, under laser irradiation, AuNPs can amplify the electromagnetic field to form submicron-sized cavitation bubbles [63,64,65]. These bubbles, which are similar to mini bombs, induce very high localized vapor pressure and disrupt the cytoskeleton of the cell. This eventually leads to membrane lysis; therefore, a lack of membrane integrity contributes significantly to cell death and is a major cause of the photothermal ablation of cells. Our results depicting the formation of such bubbles coincide well with these conclusions. Similar to our results, Rau et al. [66] reported the membrane lysis of MG63 osteoblast-like cells and the formation of such bubbles due to laser (532 nm wavelength) irradiation at 100 mW/cm^2^ intensity for 1 min.

When surface-engineered AuNPs with different morphologies (size, shape) and surface characteristics enter a biological system, their behavior (including cellular uptake, biodistribution, immune response, and toxicity) differs due to variations in their interactions with proteins and other cellular components [67,68,69]. Therefore, it is critical to perform mechanistic studies to gain a fundamental understanding of nano–bio interactions to develop safe AuNPs for biological applications. Recently, Nguyen et al. [70] highlighted the novel techniques currently used in nanoparticle–biomolecule interaction analysis and the various omics (transcriptomics, proteomics, metabolomics, epigenomics) studies to acquire mechanistic insights into engineered gold nanomaterials with different characteristics in cells. Jawaid et al. [71] reported the mechanisms of cell death induced by AuNPs of various sizes (2, 40, and 100 nm) in human myelomonocytic lymphoma (U937) cells. The results showed that the AuNPs conjugated with helium-based cold atmospheric plasma (He-CAP) induced cell death by apoptosis through the formation of reactive oxygen species (ROS). Mechanistically, the small AuNPs (2 nm) caused more significant cell death than the larger AuNPs by depleting intracellular glutathione, leading to the formation of intracellular ROS, such as superoxide (O_2_^•−^) and hydroxyl radicals (^•^OH). Similarly, Lee et al. [72] reported that AuNPs induced DNA fragmentation, resulting in an apoptosis-like cell death mechanism in *Escherichia coli*, using the terminal deoxynucleotidyl transferase (TdT) dUTP nick-end labeling (*TUNEL*) assay. Ghita et al. [73] performed a mechanistic study of AuNPs’ (1.9 nm) radiosensitization using soft X-ray (carbon K-shell, 278 eV) microbeam irradiation on MDA-MB-231 breast cancer and AG01522 fibroblast cells. The extent of DNA damage was measured using fluorescence microscopy analysis of 53BP1 (p53-binding protein 1), which is a major key protein that can recognize and repair the damaged DNA and help to maintain genome stability. The results showed differential effects of AuNPs on DNA damage and mitochondrial depolarization between the two cell lines, indicating that this effect is cell-line-specific.

Xie et al. [74] evaluated the shape effect of AuNPs in cellular uptake by treating AuNPs of different anisotropic geometries with RAW264.7 cells, which are mouse leukemic monocyte macrophages. After 24 h of incubation, the intracellular concentration of gold was quantified using inductively coupled plasma atomic emission spectrometry (ICP-AES), and the cellular uptake percentages of gold nanostars, nanorods, and nanotriangles were 0.38%, 2.04%, and 3.33%, respectively. TEM images showed that all three differently shaped AuNP types were localized in vacuoles (i.e., endosomes and/or lysosomes) in the perinuclear region of the cells. The cellular uptake mechanism, assessed using endocytic inhibitors, indicated that all of these particles entered the cells using the clathrin-mediated endocytic pathway. The interaction of AuNPs with proteins and the subsequent protein corona formation can lead to alterations in protein configuration, which, in turn, can alter their properties and functions. Nandakumar et al. [75] analyzed the human plasma protein corona formation on spherical and spiky AuNPs using label-free mass spectrometry. The experimental data indicated that the size and structure of the protein corona matched the AuNPs’ morphology. The small globular proteins and large fibrillar proteins were found predominantly on spiky AuNPs, while large proteins were adsorbed strongly on spherical AuNPs. Reports have also shown that the enhanced interaction of AuNPs with cellular systems can induce oxidative stress by producing or modulating the formation of reactive oxygen species, which is a major factor in determining their cytotoxicity. In summary, the insights obtained by the mechanistic studies of AuNPs in cells will enable researchers to design safe AuNPs with improved therapeutic efficiency for various biomedical applications.

## 4. Conclusions

The present study demonstrates the impact of ultrafast fs laser-activated photothermia using ultrasmall plasmonic AuNPs on the viability and morphology of HepG2 cells. When these cells were simply incubated with the AuNPs and not exposed to radiation, there was no notable deviation in cell viability, confirming the high biocompatibility and low cytotoxicity of AuNPs. Conversely, when these cells treated with AuNPs were exposed to 343 nm laser light with 0.1 W/cm^2^ intensity for 5 and 10 min, the cell viability was significantly reduced. These results indicate that the AuNPs plus low-intensity fs lasers produced a synergistic effect and promoted significant cell death compared to the untreated and non-irradiated control cells and the cells treated with AuNPs but not irradiated. In conclusion, this study exhibits a rapid and efficient PTT approach using ultrafast fs lasers and hydrophobic AuNPs that can be incorporated inside cell membranes to produce a synergistic photothermal effect on cancer cells. These results lay the foundation for the development of advanced cancer therapy approaches using lasers. Further studies focusing on the influence of fs lasers on the biochemical pathways, changes in enzymatic cascades, and cell death mechanisms of cancer cells will be required to gain in-depth knowledge in this area. Our future research aims to unravel these challenges and to explore AuNPs with different shapes, such as nanorods, nanostars, and nanobipyramids, to gain a better understanding of the effects of AuNP shapes and fs lasers on photothermal cancer therapy.

## Figures and Tables

**Figure 1 cells-13-02139-f001:**
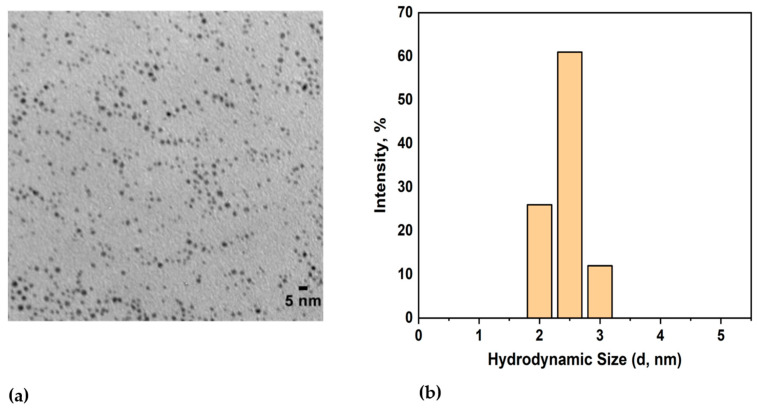
(**a**) TEM image of dodecanethiol-stabilized hydrophobic AuNPs; (**b**) DLS measurements showing the hydrodynamic diameter of AuNPs.

**Figure 2 cells-13-02139-f002:**
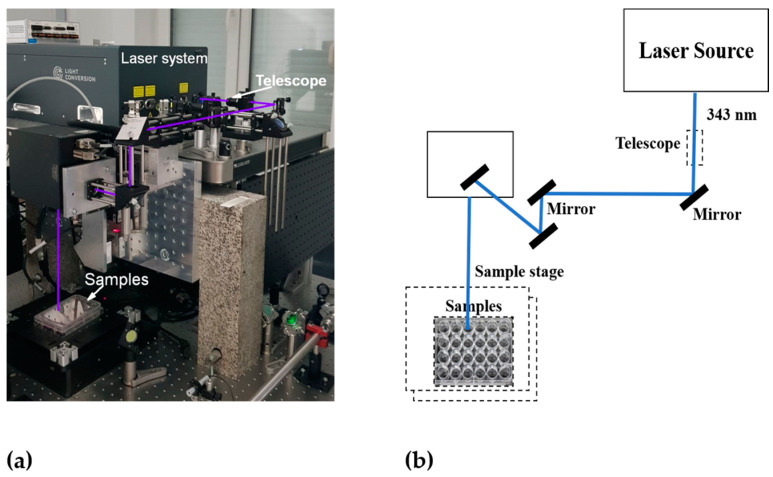
Experimental setup of the femtosecond laser system: (**a**) photo and (**b**) schematics.

**Figure 3 cells-13-02139-f003:**
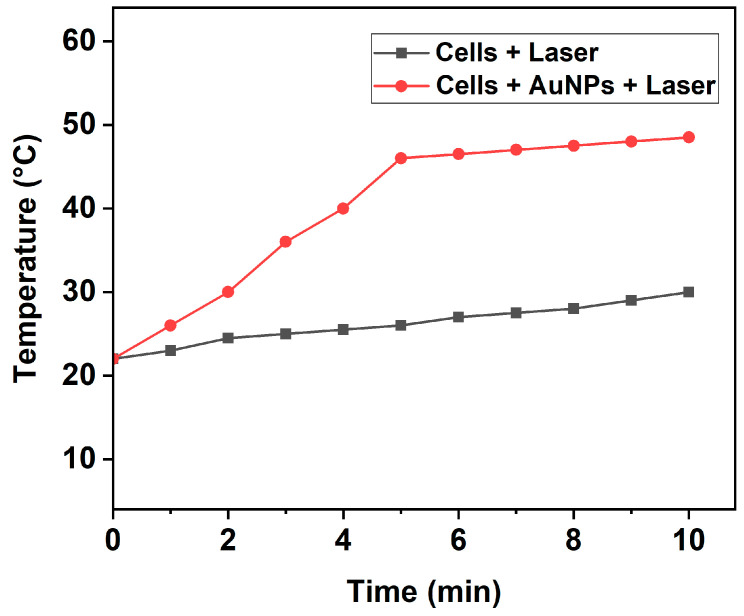
Temperature increase induced by pulsed fs laser in wells containing cells with or without AuNPs at various time intervals.

**Figure 4 cells-13-02139-f004:**
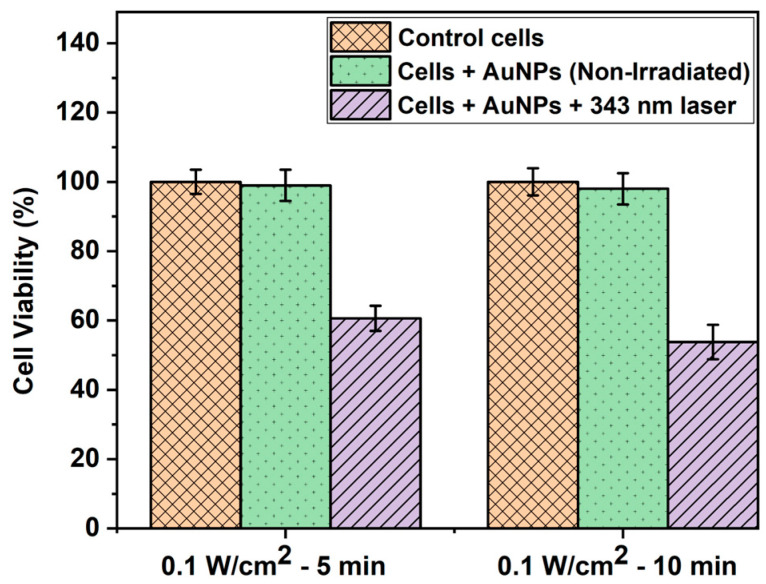
WST-1 assay result depicting the HepG2 cell viability (%) of control cells, cells treated with AuNPs alone, and cells treated with AuNPs and irradiated with a 343 nm pulsed laser with 0.1 W/cm^2^ power density at different time intervals.

**Figure 5 cells-13-02139-f005:**
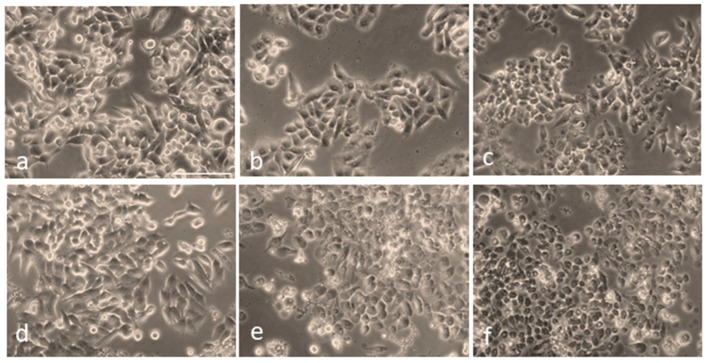
Overall morphology of HepG2 cells following irradiation at a 343 nm wavelength, with and without AuNP treatment: (**a**) Control cells (untreated, non-irradiated). (**b**) Cells irradiated at 0.1 W/cm^2^ power density for 5 min without AuNPs. (**c**) Cells irradiated for 10 min without AuNPs. (**d**) Cells treated with AuNPs but not irradiated. (**e**) Cells treated with AuNPs and irradiated with a 343 nm laser at 0.1 W/cm^2^ power density for 5 min. (**f**) Cells treated with AuNPs and irradiated under the same conditions for 10 min. Scale bar: 100 µm.

## Data Availability

The data presented in this study are available upon request from the corresponding author.

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
