# Peer review of "Femtosecond Laser-Induced Photothermal Effects of Ultrasmall Plasmonic Gold Nanoparticles on the Viability of Human Hepatocellular Carcinoma HepG2 Cells"

_cells, 2024, doi:10.3390/cells13242139_

Round 1
Reviewer 1 Report
Comments and Suggestions for Authors
Manuscript No.: cells-3340991
Manuscript title: Femtosecond Laser-induced Photothermal Effects of Ultrasmall plasmonic Gold Nanoparticles on the Viability of Human Hepatocellular Carcinoma HepG2 cells
Authors: Poornima Budime Santhosh, Kamelia Hristova-Panusheva, Todor Petrov, Lyubomir Stoychev, Natalia Krasteva, Julia Genova
Critical questions which should be answered in major revision:
1. In Materials and Methods – Cell Culture subsection – Did authors analyze cellular uptake of AuNPs after 30 minutes before exposure to laser light? 2. In Materials and Methods – Cell Culture subsection – Did Authors verify the effect of pulsed laser irradiation on selected cell lines without AuNPs? What about positive control? 3. In Results and discussion – Cell Viability Assays subsection (page 7, lines 270-271) – Authors stated that: “As seen from Figure 5a, the control cells remained intact and their proliferation pattern was normal” Did Authors determined DT (doubling time)? 4. In Results and discussion – Cell Viability Assays subsection – Figure 5 please change the figure (better quality; higher magnification), due to very low quality of microphotographs changes in cell morphology are hardly visible.
Critical questions which should be answered in minor revision:
1. In Materials and Methods – Cell Culture subsection – information about vendors/manufactures should be added or standardized “Sigma-Aldrich…” (page 3, line 122) 2. In Materials and Methods – Cell Culture subsection – lack information about format of cell culture plates, please add information “24-well plate” (page 3, lines 122-123) 3. In Materials and Methods – There are some formatting errors - the lack of spaces between words (page 3, lines 126). Please carefully check and modify.
4. In Results and discussion – Laser-induced Photothermal Effects of AuNPs subsection – some formatting errors – same phases are written in italics (page 5, line 195; page 6, line 230). Please carefully check and modify.
Author Response
Response to Reviewer 1
We thank the reviewers for their valuable feedback and appreciate their efforts to improve the manuscript. We accept their comments and prepared a point-by-point response to their comments in the revised the manuscript. The changes are highlighted in red color in the revised manuscript for easy tracking.
Reviewer 1
Comment 1: Did authors analyze cellular uptake of AuNPs after 30 minutes before exposure to laser light?
Reply: Thank you for your question. In our previous studies, we have evaluated the cellular uptake of these gold nanoparticles using the same cell lines under similar conditions at various time intervals. The experimental results have shown adequate nanoparticles uptake by cells, that is sufficient enough to induce photothermia to kill cancer cells. Since the experimental conditions were similar, we did not repeat the experiment, and assumed a similar effect. However, we acknowledge that assessing nanoparticles uptake more specifically at this time point could provide valuable insights, and will consider including such analysis in our future works.
Comment 2: In Materials and Methods – Cell Culture subsection – Did Authors verify the effect of pulsed laser irradiation on selected cell lines without AuNPs? What about positive control?
Reply: Yes, we have verified the effect of pulsed laser irradiation on the selected cell lines without AuNPs. We have included the corresponding micrographs in Figure 5 (page: 9; line: 327) in the revised manuscript to address this question. Regarding the controls, we implemented multiple experimental controls to ensure the validity of our results.
Negative Control: Cells neither treated with AuNPs nor exposed to laser irradiation;
NPs-only Control: Cells treated with AuNPs but not irradiated;
Irradiation-only Control: Cells not treated with AuNPs but exposed to laser irradiation at 343 nm for 5 and 10 minutes with a power density of 0.1 W/cm².
These controls allowed us to confirm that the observed effects on cell viability and morphology were specifically due to the synergistic interaction between AuNPs and laser irradiation. We have also added this information in Materials and Methods section 2.2. (page: 3; line: 125-132) We hope this additional clarification addresses your concern. Thank you for this valuable question.
Comment 3: In Results and discussion – Cell Viability Assays subsection (page 7, lines 270-271) – Authors stated that: “As seen from Figure 5a, the control cells remained intact and their proliferation pattern was normal” Did Authors determined DT (doubling time)?
Reply: No, we did not determine the doubling time (DT) of the control cells in this study. Our assessment of cell proliferation was based on the qualitative observation of cell morphology, and viability patterns under the experimental conditions. However, we agree that determining DT could provide additional insights and will consider incorporating this analysis in our future experiments.
Comment 4: In Results and discussion – Cell Viability Assays subsection – Figure 5 please change the figure (better quality; higher magnification), due to very low quality of microphotographs changes in cell morphology are hardly visible.
Reply: Thank you for your feedback. We appreciate your suggestion regarding Figure 5. We have replaced the micrographs, and the updated figure is included in the revised manuscript. (page: 9; line: 327)
Critical questions which should be answered in minor revision:
Comment 5: In Materials and Methods – Cell Culture subsection – information about vendors/manufactures should be added or standardized “Sigma-Aldrich…” (page 3, line 122)
Reply: The Neubauer hemocytometer’s manufacturer information is added on page: 3; line: 122.
Comment 6: In Materials and Methods – Cell Culture subsection – lack information about format of cell culture plates, please add information “24-well plate” (page 3, lines 122-123) 3.
Reply: The format of cell culture plates “24-well plate” is added on page: 3; line: 123.
Comment 7: In Materials and Methods – There are some formatting errors - the lack of spaces between words (page 3, lines 126). Please carefully check and modify.
Reply: We have corrected the formatting errors on page: 3; line: 135.
Comment 8: In Results and discussion – Laser-induced Photothermal Effects of AuNPs subsection – some formatting errors – same phases are written in italics (page 5, line 195; page 6, line 230). Please carefully check and modify.
Reply: We have corrected the formatting errors (page: 5; line: 206; page: 6; line: 241).
Reviewer 2 Report
Comments and Suggestions for Authors
Current manuscript from Poornima et al has beer reviewed, investigates the impact of 2.2 ± 1.1 nm dodecanethiol-stabilized gold nanoparticles (AuNPs) combined with femtosecond (fs) laser irradiation (343 nm wavelength, 100 mW/cm²) on HepG2 cells. Results demonstrate that the treatment reduced cell viability by 39.4% (60.6% viability remaining) after 5 minutes and by 46.2% (53.8% viability remaining) after 10 minutes. These data underscore the efficacy of the synergistic approach in inducing targeted cell death.
Check TEM image, did you publish this image before? Add new image and DLS. Why volume image added not intensity?
Due to small size 3.5 ± 1.6 nm using dynamic light scattering (DLS), while their high zeta potential was measured at −14.07 ± 0.08 mV these nanoparticles are not useful for any in vivo experiment. They will be excreted out. What’s the use of study then?
Its good 100 mW/cm² used ensuring minimal damage to surrounding healthy tissues. The chosen wavelength of 343 nm corresponded to the absorption peak of the AuNPs, optimizing the photothermal conversion efficiency. What is the power in this case to create most damage to cells? Add more results or explain it. Compare results with Gold nanoords, check Prof. Mostafa A. El-Sayed papers, 10.1021/acsnano.8b01424, 10.1080/14686996.2016.1189797, etc.
In summary, this study achieved a significant reduction in cell viability, a controlled temperature increase, and targeted apoptosis using ultrasmall AuNPs and fs laser irradiation. The findings highlight a potential pathway to develop safer, more effective cancer therapies with reduced side effects and improved precision. I recommend after applying above corrections.
Author Response
Response to Reviewers
We thank the reviewers for their valuable feedback and appreciate their efforts to improve the manuscript. We accept their comments and prepared a point-by-point response to their comments in the revised the manuscript. The changes are highlighted in red color in the revised manuscript for easy tracking.
Reviewer 2
Current manuscript from Poornima et al has been reviewed, investigates the impact of 2.2 ± 1.1 nm dodecanethiol-stabilized gold nanoparticles (AuNPs) combined with femtosecond (fs) laser irradiation (343 nm wavelength, 100 mW/cm²) on HepG2 cells. Results demonstrate that the treatment reduced cell viability by 39.4% (60.6% viability remaining) after 5 minutes and by 46.2% (53.8% viability remaining) after 10 minutes. These data underscore the efficacy of the synergistic approach in inducing targeted cell death.
Comment 1: Check TEM image, did you publish this image before? Add new image and DLS. Why volume image added not intensity?
Reply: We have added a new TEM image (Figure 1a) and a DLS image (Figure 1b) based on intensity, as suggested by the reviewer, on page No. 4, line number 176.
Comment 2: Due to small size 3.5 ± 1.6 nm using dynamic light scattering (DLS), while their high zeta potential was measured at −14.07 ± 0.08 mV these nanoparticles are not useful for any in vivo experiment. They will be excreted out. What’s the use of study then?
Reply: Thank you for your comment. We understand the limitations of using small-sized AuNPs with a negative zeta potential for in vivo applications due to their potential rapid clearance. However, our study focuses on understanding their interactions with cells and their effects in vitro. The findings provide valuable insights into the cellular responses and mechanisms of action associated with these nanoparticles, which can inform the design of future nanoparticle systems optimized for in vivo use. Additionally, the study contributes to the broader understanding of nanoparticles behavior and cytotoxicity, which is critical for advancing nanotechnology in biomedical applications. We appreciate your concern and consider these factors while synthesizing AuNPs for in vivo applications in our future work.
Comment 3: Its good 100 mW/cm² used ensuring minimal damage to surrounding healthy tissues. The chosen wavelength of 343 nm corresponded to the absorption peak of the AuNPs, optimizing the photothermal conversion efficiency. What is the power in this case to create most damage to cells? Add more results or explain it. Compare results with Gold nanoords, check Prof. Mostafa A. El-Sayed papers, 10.1021/acsnano.8b01424, 10.1080/14686996.2016.1189797, etc.
Reply: In our preliminary experiments, we observed that when the laser power was increased to 200 mW/cm² for 10 minutes, the photothermal effect induced by AuNPs damaged most of the cancer cells. But the associated temperature increase at this laser power was more than 50°C. As mentioned in this paper (Pg. No 6 & 7, line No: 241-253), temperature above 50 °C induces cell death via the mechanism of necrosis, which is not preferable as it can cause undesirable side effects to the healthy cells near the tumor region, probably due to inflammation. Hence, we used 100 mW/cm² laser intensity to increase the temperature less than 50 °C to induce cell death by apoptosis mechanism to minimize the damage to healthy cells. As suggested by the reviewer, we have referred the above articles, and compared the laser power required to significantly reduce cell viability of cancer cells treated with gold nanorods. The newly added data is on pages: 7 and 8; lines: 277-312, and the reference numbers of articles are updated accordingly.
Response to Reviewers
We thank the reviewers for their valuable feedback and appreciate their efforts to improve the manuscript. We accept their comments and prepared a point-by-point response to their comments in the revised the manuscript. The changes are highlighted in red color in the revised manuscript for easy tracking.
Reviewer 2
Current manuscript from Poornima et al has been reviewed, investigates the impact of 2.2 ± 1.1 nm dodecanethiol-stabilized gold nanoparticles (AuNPs) combined with femtosecond (fs) laser irradiation (343 nm wavelength, 100 mW/cm²) on HepG2 cells. Results demonstrate that the treatment reduced cell viability by 39.4% (60.6% viability remaining) after 5 minutes and by 46.2% (53.8% viability remaining) after 10 minutes. These data underscore the efficacy of the synergistic approach in inducing targeted cell death.
Comment 1: Check TEM image, did you publish this image before? Add new image and DLS. Why volume image added not intensity?
Reply: We have added a new TEM image (Figure 1a) and a DLS image (Figure 1b) based on intensity, as suggested by the reviewer, on page No. 4, line number 176.
Comment 2: Due to small size 3.5 ± 1.6 nm using dynamic light scattering (DLS), while their high zeta potential was measured at −14.07 ± 0.08 mV these nanoparticles are not useful for any in vivo experiment. They will be excreted out. What’s the use of study then?
Reply: Thank you for your comment. We understand the limitations of using small-sized AuNPs with a negative zeta potential for in vivo applications due to their potential rapid clearance. However, our study focuses on understanding their interactions with cells and their effects in vitro. The findings provide valuable insights into the cellular responses and mechanisms of action associated with these nanoparticles, which can inform the design of future nanoparticle systems optimized for in vivo use. Additionally, the study contributes to the broader understanding of nanoparticles behavior and cytotoxicity, which is critical for advancing nanotechnology in biomedical applications. We appreciate your concern and consider these factors while synthesizing AuNPs for in vivo applications in our future work.
Comment 3: Its good 100 mW/cm² used ensuring minimal damage to surrounding healthy tissues. The chosen wavelength of 343 nm corresponded to the absorption peak of the AuNPs, optimizing the photothermal conversion efficiency. What is the power in this case to create most damage to cells? Add more results or explain it. Compare results with Gold nanoords, check Prof. Mostafa A. El-Sayed papers, 10.1021/acsnano.8b01424, 10.1080/14686996.2016.1189797, etc.
Reply: In our preliminary experiments, we observed that when the laser power was increased to 200 mW/cm² for 10 minutes, the photothermal effect induced by AuNPs damaged most of the cancer cells. But the associated temperature increase at this laser power was more than 50°C. As mentioned in this paper (Pg. No 6 & 7, line No: 241-253), temperature above 50 °C induces cell death via the mechanism of necrosis, which is not preferable as it can cause undesirable side effects to the healthy cells near the tumor region, probably due to inflammation. Hence, we used 100 mW/cm² laser intensity to increase the temperature less than 50 °C to induce cell death by apoptosis mechanism to minimize the damage to healthy cells. As suggested by the reviewer, we have referred the above articles, and compared the laser power required to significantly reduce cell viability of cancer cells treated with gold nanorods. The newly added data is on pages: 7 and 8; lines: 277-312, and the reference numbers of articles are updated accordingly.